# Rapid prediction of in-hospital mortality among adults with COVID-19 disease

**Kyoung Min Kim**[1,2‡], **Daniel S. Evans**[1‡], **Jessica Jacobson**[3], **Xiaqing Jiang**[4], **Warren Browner**[1], **Steven R. Cummings**[1,5]*

**1** San Francisco Coordinating Center, California Pacific Medical Center Research Institute, Sutter Health, San Francisco, California, United States of America, **2** Division of Endocrinology, Department of Internal Medicine, Yongin Severance Hospital, Yonsei University College of Medicine, Yongin, South Korea, **3** New York City Health + Hospitals/Bellevue-NYU Grossman School of Medicine, New York, New York, United States of America, **4** Orthopedic Surgery, School of Medicine, University of California San Francisco, San Francisco, California, United States of America, **5** Department of Epidemiology and Biostatistics, University of California San Francisco, San Francisco, California, United States of America

‡ KMM and DSE as co-first authors.
* steven.cummings@sfcc-cpmc.net

**Data Availability Statement:** The data file for the paper was created especially for this manuscript by extracting specified variables from NYC Health and Hospitals EPIC medical records. This original

## Abstract

### Background

We developed a simple tool to estimate the probability of dying from acute COVID-19 illness only with readily available assessments at initial admission.

### Methods

This retrospective study included 13,190 racially and ethnically diverse adults admitted to one of the New York City Health + Hospitals (NYC H+H) system for COVID-19 illness between March 1 and June 30, 2020. Demographic characteristics, simple vital signs and routine clinical laboratory tests were collected from the electronic medical records. A clinical prediction model to estimate the risk of dying during the hospitalization were developed.

### Results

Mean age (interquartile range) was 58 (45–72) years; 5421 (41%) were women, 5258 were Latinx (40%), 3805 Black (29%), 1168 White (9%), and 2959 Other (22%). During hospitalization, 2,875 were (22%) died. Using separate test and validation samples, machine learning (Gradient Boosted Decision Trees) identified eight variables—oxygen saturation, respiratory rate, systolic and diastolic blood pressures, pulse rate, blood urea nitrogen level, age and creatinine—that predicted mortality, with an area under the ROC curve (AUC) of 94%. A score based on these variables classified 5,677 (46%) as low risk (a score of 0) who had 0.8% (95% confidence interval, 0.5–1.0%) risk of dying, and 674 (5.4%) as high-risk (score ≥ 12 points) who had a 97.6% (96.5–98.8%) risk of dying; the remainder had intermediate risks. A risk calculator is available online at https://danielevanslab.shinyapps.io/Covid_mortality/.

dataset was created and transferred to the
corresponding author. The original dataset is
available from: Andres Parker, Epic Business
Intelligence Developer extracted, prepared and
provided the dataset. His contact information is:
Central Office - NYC Health + Hospitals 50 Water
Street, 5th Floor; New York, NY 10041; email
address: parkera10@nychhc.org. In addition: The
dataset used for the analyses in the paper are
ALSO available by contacting the corresponding
author, Steven Cummings, at steven.
cummings@ucsf.edu.

**Funding:** The authors received no specific funding
for this work.

**Competing interests:** The authors have declared
that no competing interests exist.

## Conclusions

In a diverse population of hospitalized patients with COVID-19 illness, a clinical prediction model using a few readily available vital signs reflecting the severity of disease may precisely predict in-hospital mortality in diverse populations and can rapidly assist decisions to prioritize admissions and intensive care.

## Introduction

Hospitals in many countries have been overwhelmed again by admissions of patients with COVID-19 illness in the second wave of delta variant infections or other types of variants. Accurate prediction of the probability of death using rapidly available vital signs on arrival or immediately after admission to hospital without further testing of laboratory parameters or chest x-ray, might help prioritize patients for hospitalization, intensive care and intubation, or to receive limited treatments in places that have very limited resources.

Several prediction models about dying from COVID-19 disease has been proposed [1], but they have several limitations when immediate applying at initial admission of patients. Previous prediction algorithms have been derived from small numbers of deaths [2–10], used comorbid conditions, diagnoses, and severity indices from electronic medical records assessed after the patient is admitted [5, 11–18], or included tests—such as levels of C-reactive protein, troponin, D-dimers—that may not be readily available for urgent triage of patients for hospital admission or intensive care [18, 19]. Some studies were done in ethnically homogenous populations such as Wuhan [20], China [2], or Italy [21], or specific populations such as nursing home residents [22] or community based registry [23]. Some studies have been done in patients already admitted to the hospital with clinical or laboratory results after admission, or patients already in an intensive care unit (ICU) [7, 11, 12] or after admission to the hospital [19] and, therefore, not applicable to features of the infection when first presenting to emergency care. Some imposed arbitrary durations of follow-up, such as 7 or 30 days. Some studies applied machine learning methods to develop predictive models [2, 6, 24, 25].

No previous model has been based on measurements immediately available at the time of triage in a large racially diverse population. No model has been translated into a calculator that can be used on mobile devices in clinical settings. The model and online calculator are likely to apply to all variants of COVID-19 infection because, although the delta variant has greater viral loads and risk of transmission, there is no evidence that the physiologic manifestations, such as hypoxia differ or that the clinical manifestations that predict mortality would differ between the variants [26] (https://www.cdc.gov/coronavirus/2019-ncov/variants/variant.htm).

We developed a predictive algorithm based on readily data from initial evaluation before admission to a hospital, in a diverse patient population, and mortality at any time after admission. We studied the large and diverse population of patients admitted to New York City Health + Hospitals (NYC H+H) public hospital system. We used machine learning to select strong predictors of mortality, developed and validated a multivariable model and score to estimate the risk of dying, and translated the model into an online calculator to estimate the risk of in-hospital mortality.

## Methods

### Setting and data sources

We used data extracted from the electronic medical records of all patients at least 18 years old who were admitted to any of the 11 hospitals of the New York City Health + Hospitals (NYC

H+H) system with a diagnosis of Covid 19 infection verified with a positive polymerase chain reaction (PCR) test between March 1 and June 30, 2020. NYC H+H is the largest public health system in the United States, providing health services to more than one million New Yorkers across the city's five boroughs. These hospitals account for approximately one-fifth of all general hospital discharges and more than one-third of emergency department and hospital-based clinic visits in New York City.

## Variables

We abstracted demographic characteristics (sex, age, race and ethnicity), weight, body mass index, vital signs, oxygen saturation (SpO2) from peripheral monitors, and routine clinical laboratory tests (serum chemistry panel, complete blood counts) and D-dimer levels from electronic medical records. When there was more than one value, we selected the first. Missing values were not imputed. Non-transformed values were used. Sex and race/ethnicity were coded as categorical variables; all others were recorded as continuous variables. (Results did not change when continuous features were centered to their mean and scaled to a standard deviation of one.) The outcome was death from any cause during hospitalization with COVID19 infection; length of hospitalization was also noted.

## Statistical analysis

The assumptions of normality for collected variables were tested Kolmogrov-Smirnov test, which is known to be sensitive in two samples. Baseline characteristics are presented median (IQR) for non-normally distributed continuous variables or N (%) for categorical variables. Because most of predictors were not distributed normally, the descriptive statistics for baseline characteristics were compared by in-hospital mortality using Mann-Whitney U test for continuous variables and Chi-squared test for categorical variables.

To develop a clinical prediction model to estimate the risk of dying in the hospital, we adopted a multistep approach that included variable selection using Extreme Gradient Boosted Decision Trees (XGBoost), followed by the identification of cut points of the selected variables using classification and regression trees (CART), then followed by the development of a score that was used to predict in-hospital mortality within Covid-19 positive patients in the study population. Train and test data partitions were created using an 80%/20% random split stratified by death status to ensure an even proportion of mortality in the train and test partitions. Gradient Boosted Decision Trees implemented in the XGBoost R package v 1.2.0.1 with R v 4.0.2. were used to generate an ensemble of multiple decision trees to minimize errors in the classification of mortality in patients. The XGBoost model was developed in the train partition, using four boosting rounds, a maximum depth of three for each decision tree, a learning rate of 0.3, a binary: logistic learning objective with error rate used as the evaluation metric, and a minimum child weight of 75. Variable importance was evaluated using the information gain metric of a split on a variable. XGBoost model performance was evaluated in the test partition using accuracy and area under the curve (AUC) from a receiver operating characteristic (ROC) curve. Selected features and model performance did not change with 10-fold cross-validation.

To develop a clinical prediction score, we used Classification and Regression Tree (CART) analyses in the original training set to identify optimum cut-points for each variable selected by XGBoost (S1 Fig). There was no clear cut-point for creatinine level and it had low importance in the XGBoost model, therefore it was not included in the final calculation of clinical risk score. We entered the selected variables and cut-points into a logistic regression model to estimate the multivariable odds ratios. To assign risk scores, the odds ratio for each of these

categorical variables were divided by 2.6 (the lowest odds ratio), rounded, and then summed for each patient to calculate a risk score. The risk score calculation was not changed after including categories for missing values for all selected variables. The predicted probability of mortality from the risk score was also compared with the observed mortality in the test set.

After excluding 703 patients with missing values for one or more variables, the proportions of patients who died were calculated for each 1-point interval in risk score; the highest-risk categories, which had similar scores and small numbers of patients, were combined. Because the predicted mortality by risk score categories were very similar in the training and test sets (AUC = 0.94 for both), these sets were combined to estimate the probabilities and 95% confidence intervals for the entire population. An online calculator reports the probability of in-hospital mortality from the risk score (danielevanslab.shinyapps.io/Covid_mortality). To report the probability of dying, all variables must have non-missing values except for the blood urea nitrogen (BUN) test which includes a term for missing results.

All statistical analysis was performed using R Statistical Software (version 4.0.3 and version 4.0.2; R Foundation for Statistical Computing, Vienna, Austria).

### Patient and public involvement

Patients or members of the public were not included in the analysis owing to restriction on the use of the data included in the study and a lack of training in the use of these data.

### Results

Between March 1 and June 30, 2020, 13,190 patients who confirmed with COVID-19 infection, were admitted to a NYC H+H hospital. Among them, 2,227 (16.9%) patients were cared in ICU during hospitalization. The cohort included 5421 [41.1%] women, mean age 58 years [interquartile range 45–72 years]; 5258 were Hispanic [39.9%], 3805 Black [28.8%], 1168 White [8.9%], 716 Asian [5.4%] and 2243 individuals of other races/ethnicities [17.0%] (Table 1). During hospitalization, 2,875 (21.8%) died a mean of 10.6 days after admission (interquartile range: 3 to 13 days) and 2279 (17.3) were treated with mechanical ventilation.

There were statistically significant differences between those who died and those who survived for almost all variables (Table 1). The XGBoost algorithm identified eight variables (Fig 1) that, together, generated predictions of mortality with an AUC of 94% and an accuracy of 91% (Fig 2). Of the variables that the XGBoost model selected, SpO2 was the strongest predictor; respiratory rate and blood pressure were also major contributors; body temperature was not. Although race and ethnicity were associated with mortality in univariable analyses, they were not selected in the predictive model.

CART analysis identified cut-points for each of the XGBoost-selected variables. A multivariable logistic model showed that the selected cut-points were all significant predictors of mortality (Table 2). The risk score based on the odds ratios for these variables ranged from 0 to 22 points and had an AUC of 0.94 for predicting mortality, the same as the XGBoost algorithm (Fig 2). The calibration curve of the risk score on the test set also showed excellence predictability over the full range probabilities of mortality (slope = 1, Brier score 0.061, Fig 3).

Among the total study subjects, there were 5,677 (45.5%) patients with a score of 0, and 674 (5.4%) with a score $\geq$ 12 points (Table 3). In-hospital mortality increased continuously with higher risk scores, ranging from ranged from 0.8% (95% confidence interval, 0.5–1.0%) for those with a score of 0 to 97.6% (96.5–98.8%) for patients with a score $\geq$ 12 points (Table 3). The mean times between admission and death was 18 days (IOR 6–27 days) for those with a risk score of 0, compared with 9 days (IQR 3–11 days) for those with a risk score of 12 or greater. We translated the models into an online calculator to report the probability of

**Table 1. Baseline characteristics of the patient population and of those who survived and died.**

| | Total | Survived | Died | p-value |
|---|---|---|---|---|
| | N = 13,190 | N = 10,315 | N = 2,875 | |
| Demographics and diagnoses | | | | |
| Age (years) | 59.0 (45.0, 72.0) | 55.0 (42.0, 67.0) | 72.0 (61.0, 81.0) | <0.001 |
| BMI (kg/m2) | 28.1 (24.4, 32.60) | 28.1 (24.4, 32.5) | 28.4 (24.5, 33.0) | 0.032 |
| Female (%) | 5421 (41.1) | 4320 (41.9) | 1101 (38.3) | 0.001 |
| Hypertension (%) | 6552 (49.7) | 4804 (46.6) | 1748 (60.8) | <0.001 |
| Diabetes (%) | 4635 (35.1) | 3339 (32.4) | 1296 (45.1) | <0.001 |
| Race/Ethnicity (%) | | | | <0.001 |
| American Indian or Alaskan | 22 (0.2) | 17 (0.2) | 5 (0.2) | |
| Asian | 716 (5.4) | 547 (5.3) | 169 (5.9) | |
| Black | 3805 (28.8) | 2962 (28.7) | 843 (29.3) | |
| White | 1168 (8.9) | 808 (7.8) | 360 (12.5) | |
| Pacific Islander | 8 (0.1) | 6 (0.1) | 2 (0.1) | |
| Hispanic | 5258 (39.9) | 4289 (41.6) | 969 (33.7) | |
| Other | 1601 (12.1) | 1246 (12.1) | 355 (12.3) | |
| Declined/Unknown | 612 (4.6) | 440 (4.3) | 172 (6.0) | |
| Vital signs | | | | |
| O2 Saturation (%) | 97.0 (95.0, 98.0) | 97.0 (95.0, 99.0) | 91.5 (80.0, 96.0) | <0.001 |
| Body temperature (°F) | 98.5 (97.9, 99.0) | 98,4 (98.0, 98.9) | 91.5 (80.0, 96.0) | 0.015 |
| Pulse rate (/min) | 85.0 (74.0, 97.0) | 85.0 (75.0, 94.0) | 89.0 (60.0, 109.0) | <0.001 |
| Respiratory rate (/min) | 18.0 (18.0, 20.0) | 18.0 (18.0, 20.0) | 20.0 (18.0, 26.3) | <0.001 |
| Systolic BP (mmHg) | 121.0 (108.0, 134.0) | 124.0 (112.0, 136.0) | 102.0 (76.0, 126.0) | <0.001 |
| Diastolic BP (mmHg) | 72.0 (63.0, 80.0) | 74.0 (67.0, 81.0) | 56.0 (41.0, 70.0) | <0.001 |
| Laboratory parameters | | | | |
| Calcium (mg/dL) | 8.4 (4.9, 9.0) | 8.5 (5.2, 9.1) | 8.1 (4.7, 8.7) | <0.001 |
| Glucose (mg/dL) | 125.0 (104.0, 178.0) | 119.0 (101.0, 159.8) | 153.0 (118.0, 232.0) | <0.001 |
| BUN (mg/dL) | 16.0 (11.0, 29.0) | 14.0 (10.0, 22.5) | 29.0 (17.0, 53.0) | <0.001 |
| Creatinine (mg/dL) | 1..0 (0.8, 1.5) | 1.0 (0.8, 1.3) | 1.4 (1.0, 2.5) | <0.001 |
| Albumin (mg/dL) | 3.7 (3.1, 4.1) | 3.8 (3.2, 4.2) | 3.4 (2.7, 3.8) | <0.001 |
| Magnesium (mg/dL) | 2.1 (1.9, 2.4) | 2.1 (1.8, 2.3) | 2.2 (1.9, 2.5) | <0.001 |
| Sodium (mg/dL) | 137.0 (134.0, 140.0) | 137.0 (134.0, 140.0) | 137.0 (133.0, 142.0) | <0.001 |
| Potassium (mg/dL) | 4.2 (3.8, 4.6) | 4.2 (3.8, 4.6) | 4.3 (3.9, 4.9) | <0.001 |
| Chloride (mg/dL) | 100.0 (96.0, 104.0) | 100.0 (96.0, 103.0) | 100.0 (95.0, 106.0) | <0.001 |
| CO2 (mg/dL) | 23.0 (20.0, 25.0) | 23.0 (21.0, 25.7) | 21.0 (18.0, 24.0) | <0.001 |
| Anion Gap | 15.6 (13.0, 18.0) | 15.0 (13.0, 17.0) | 18.0 (15.0, 21.0) | <0.001 |
| White blood cells (x $10^9$/L) | 7.6 (5.6, 10.5) | 7.2 (5.4, 9.9) | 8.9 (6.4, 12.3) | <0.001 |
| Red cell distribution width (%) | 13.6 (12.8, 14.8) | 13.4 (12.7, 14.6) | 14.2 (13.2, 15.7) | <0.001 |
| Red blood cell count (x $10^9$/L) | 4.6 (4.1, 5.0) | 4.6 (4.1, 5.0) | 4.4 (3.9, 5.0) | <0.001 |
| Hemoglobin (g/L) | 13.0 (11.5, 14.3) | 13.1 (11.7, 14.4) | 12.7 (10.8, 14.2) | <0.001 |
| Hematocrit (L/L) | 39.9 (35.8, 43.6) | 40.10 (36.0, 43.0) | 39.30 (34.0, 43.0) | <0.001 |
| Mean corpuscular volume (fL) | 88.2 (84.3, 92.0) | 87.90 (84.2, 91.5) | 89.20 (84.9, 93.4) | <0.001 |
| Mean corpuscular hemoglobin (pg) | 28.90 (27.2, 30.2) | 28.90 (27.3, 30.2) | 28.80 (27.1, 30.2) | 0.226 |
| MCHC (g/L) | 32.50 (31.5, 33.5) | 32.70 (31.7, 33.60) | 32.10 (30.9, 33.2) | <0.001 |
| Platelets (x 109/L) | 216.0 (167.0, 278.0) | 218.00 (171.0, 279.0) | 207.0 (153.0, 274.0) | <0.001 |
| Mean platelet volume (fL) | 10.7 (10.0, 11.5) | 10.6 (9.9, 11.4) | 10.9 (10.2, 11.7) | <0.001 |
| Basophil (%) | 0.2 (0.1, 0.3) | 0.2 (0.1, 0.3) | 0.2 (0.1, 0.3) | <0.001 |
| Immature granulocyte (%) | 0.19 (0.04, 0.50) | 0.15 (0.03, 0.50) | 0.30 (0.06, 0.73) | <0.001 |

(*Continued*)

**Table 1.** (Continued)

| | Total | Survived | Died | p-value |
|---|---|---|---|---|
| | **N = 13,190** | **N = 10,315** | **N = 2,875** | |
| Neutrophils (x 109/L) | 5.65 (3.82, 8.41) | 5.24 (3.62, 7.73) | 7.10 (4.87, 10.33) | <0.001 |
| Lymphocytes (x 109/L) | 1.05 (0.74, 1.48) | 1.12 (0.80, 1.55) | 0.87 (0.61, 1.23) | 0.003 |
| Monocytes (x 109/L) | 0.50 (0.35, 0.71) | 0.51 (0.36, 0.72) | 0.47 (0.31, 0.70) | 0.445 |
| Eosinophils (x 109/L) | 0.01 (0.00, 0.04) | 0.01 (0.00, 0.05) | 0.00 (0.00, 0.02) | <0.001 |
| Nucleated red blood cell (/uL)) | 0.00 (0.00, 0.00) | 0.00 (0.00, 0.00) | 0.00 (0.00, 0.02) | <0.001 |
| International normalized ratio (INR) | 1.2 (1.1, 1.3) | 1.1 (1.1, 1.2) | 1.2 (1.1, 1.4) | <0.001 |
| D-Dimer (ng/mL) | 594 (324, 1,644) | 500 (287, 1,062) | 925 (463, 3,224) | <0.001 |

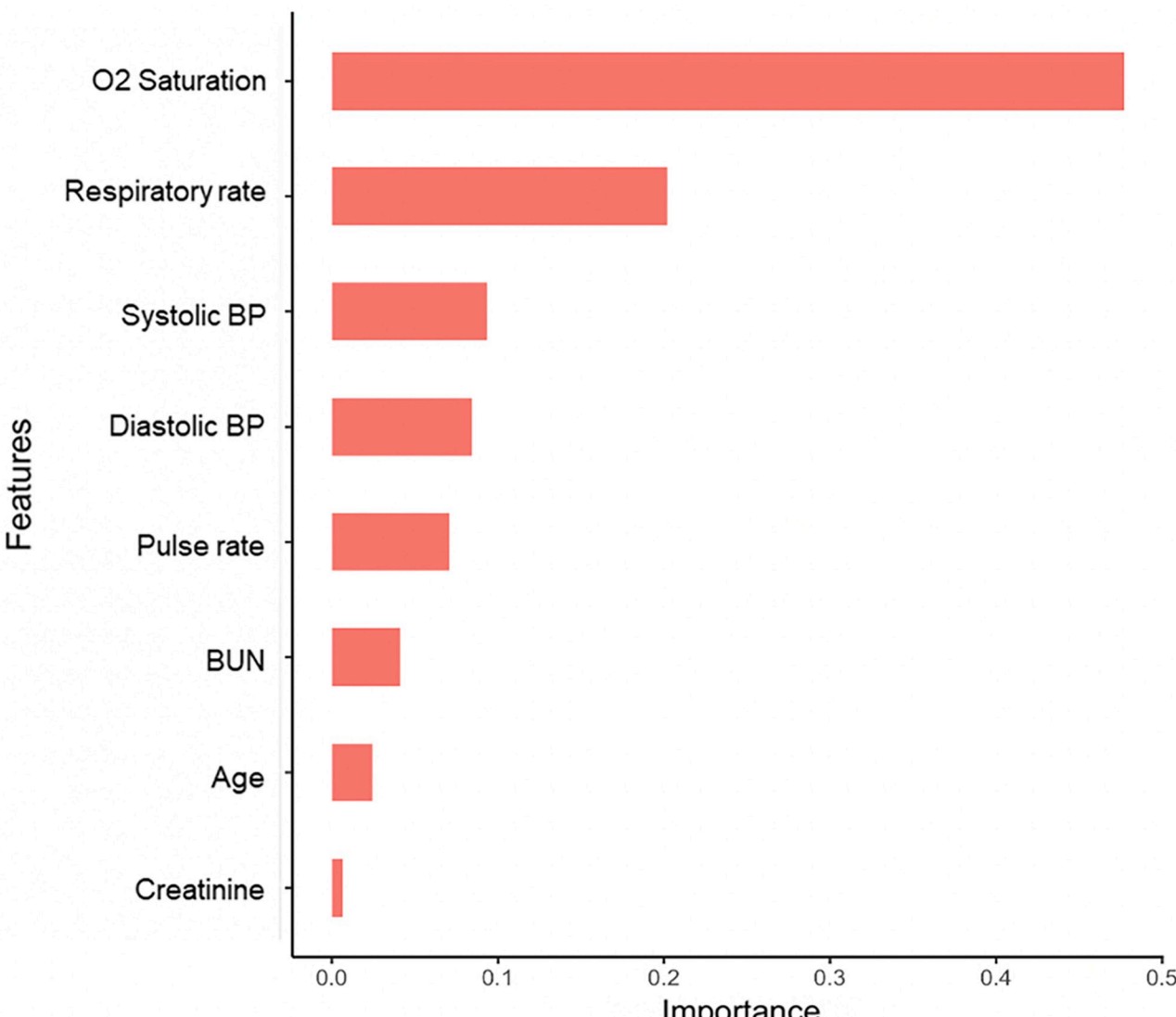

**Fig 1. Features, or variables, identified by the XGBoost model and ranked by importance, based on the gain in the accuracy of classification when the variable was used in decision trees that generated the model.**

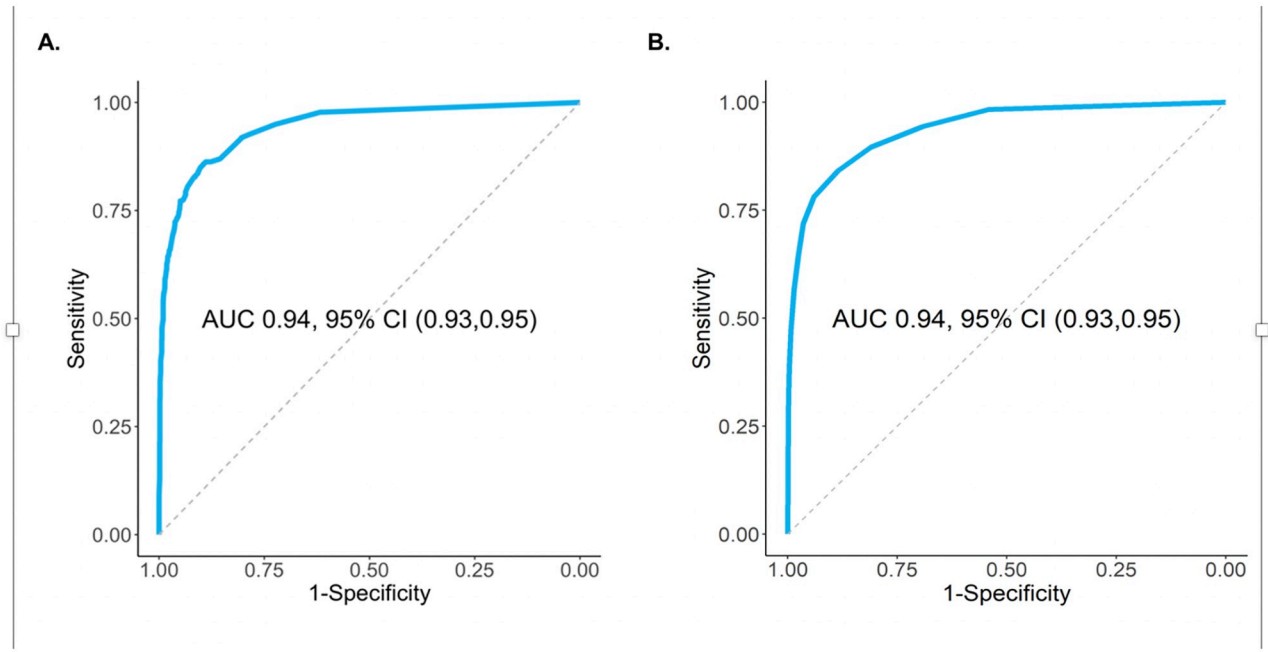

**Fig 2. Receiver Operating Characteristics (ROC) Curves of mortality predicted by the machine learning XGBoost model and the clinical prediction model based on total point score in the test set of data.** A. ROC for In-hospital Mortality Predicted by the Machine Learning XGBoost Model. B. ROC Curve for the Clinical Prediction Model Point Score.

mortality and the corresponding 95% confidence interval: danielevanslab.shinyapps.io/COV-ID_mortality/.

## Discussion

A few clinical observations readily available in the initial assessment of patients with COVID-19 infection can estimate the probability of dying during hospitalization across a full spectrum of outcomes. The model is available online for convenient use in acute care settings.

Not surprisingly, physiologic variables, such as $SpO_2$, respiratory rate, and low blood pressures were important predictors, indicating that the pulmonary and systemic effects of the infection are its most important prognostic features. Both slow and rapid respiratory rates and slow and fast pulse rates indicated an increased risk of in-hospital mortality. As expected, mortality also increased with age and with higher BUN levels [6, 27, 28]. Notably, after considering other variables, race and ethnicity were not significant predictors of mortality, as has been seen in other studies [9, 14, 29].

Previous studies have had important limitations, particularly studying patients who were already admitted and including assessments that are generally not available at the time the decision is made whether to admit a patient to the hospital [5, 11–18]. Most studies have developed models for predicting mortality from COVID-19 infections that are less accurate than the one presented here. For example, one study, applied XG boost to select variables from hospital admission in UK hospitals to from which a validated 4C Mortality model generated an AUC = 0.74 that was better than 17 other models with which it was compared, but not as accurate as the model we developed [30]. It included laboratory tests (c-reactive protein and urea) and number of comorbidities and Glasgow coma score that may require the medical record and neurological exam [30]. Other studies have identified other laboratory values, such as red

**Table 2. Predictors from the multivariable model and points indicating an increased risk of death.**

| | Odds ratio (95% CI) | Points |
|---|---|---|
| *Age* | | |
| Age < 70 years old | Reference | 0 |
| 70 ≤ Age < 85 years old | 2.6 (2.2–3.1) | 1 |
| Age ≥ 85 years old | 5.4 (4.2–7.0) | 2 |
| *O2 Saturation (SpO2)* | | |
| SpO2 ≥ 91% | Reference | 0 |
| SpO2 < 91% | 10.7 (8.3–19.9) | 4 |
| *Respiratory Rate (RR)* | | |
| 14 ≤ RR<22/min | Reference | 0 |
| RR ≥ 22/min | 7.8 (4.2–14.4) | 3 |
| RR < 14/min | 9.2 (7.6–11.1) | 4 |
| *Pulse Rate (PR)* | | |
| 51 ≤ PR < 109/min | Reference | 0 |
| PR < 51/min | 3.5 (2.6–4.7) | 1 |
| 109 ≤ PR < 119/min | 9.7 (4.8–20.0) | 4 |
| PR ≥ 119/min | 12.5 (9.1–17.3) | 5 |
| *Systolic BP (SBP)* | | |
| SBP ≥ 95 mmHg | Reference | 0 |
| SBP < 95 mmHg | 7.9 (5.8–10.7) | 3 |
| *Diastolic BP (DBP)* | | |
| DBP ≥ 54 mmHg | Reference | 0 |
| DBP < 54 mmHg | 4.7 (3.6–6.1) | 2 |
| *BUN* | | |
| BUN < 20 mmHg | Reference | 0 |
| 20 ≤ BUN < 44 mmHg | 2.6 (2.1–3.2) | 1 |
| BUN ≥ 44 mmHg | 5.9 (4.8–7.4) | 2 |
| *Range of risk score* | | *0–22* |

BP, blood pressure; BUN, blood urea nitrogen.

cell distribution width and D-dimer levels, as significant predictors, but they did not contribute to this algorithm [4, 31]. BUN was the only laboratory value in our algorithm and a missing value did not influence the score and it is optional for estimating the risk using the website. This suggests that clinicians do not need to order or wait for laboratory test results to estimate a patient's probability of dying.

These data were collected before effective treatments, such as corticosteroids, which were used commonly in the treatment of COVID-19 disease [30]. Improvements in care of patients have reduced inpatient mortality from the infection [32]. Although our risk model and algorithm is not calibrated to the current mortality risk, it does reflect the probability of dying without current in-hospital treatments and, thus may be useful to identify patients who are most—or least—likely to benefit from hospital care. However, infection in people who have been fully vaccinated may be less severe and our model may therefore overestimate mortality in those uncommon cases. Studies suggest that the delta variant carries a greater risk of hospitalization [33]. However, there is no evidence that the physiologic or clinical manifestations that would relate to the risk of mortality would differ between the variants and wild type. Therefore, our model of risk of in hospital mortality is likely to apply to all variants. Ideally, prognostic models developed for the alpha variant would be recalibrated for the delta variant.

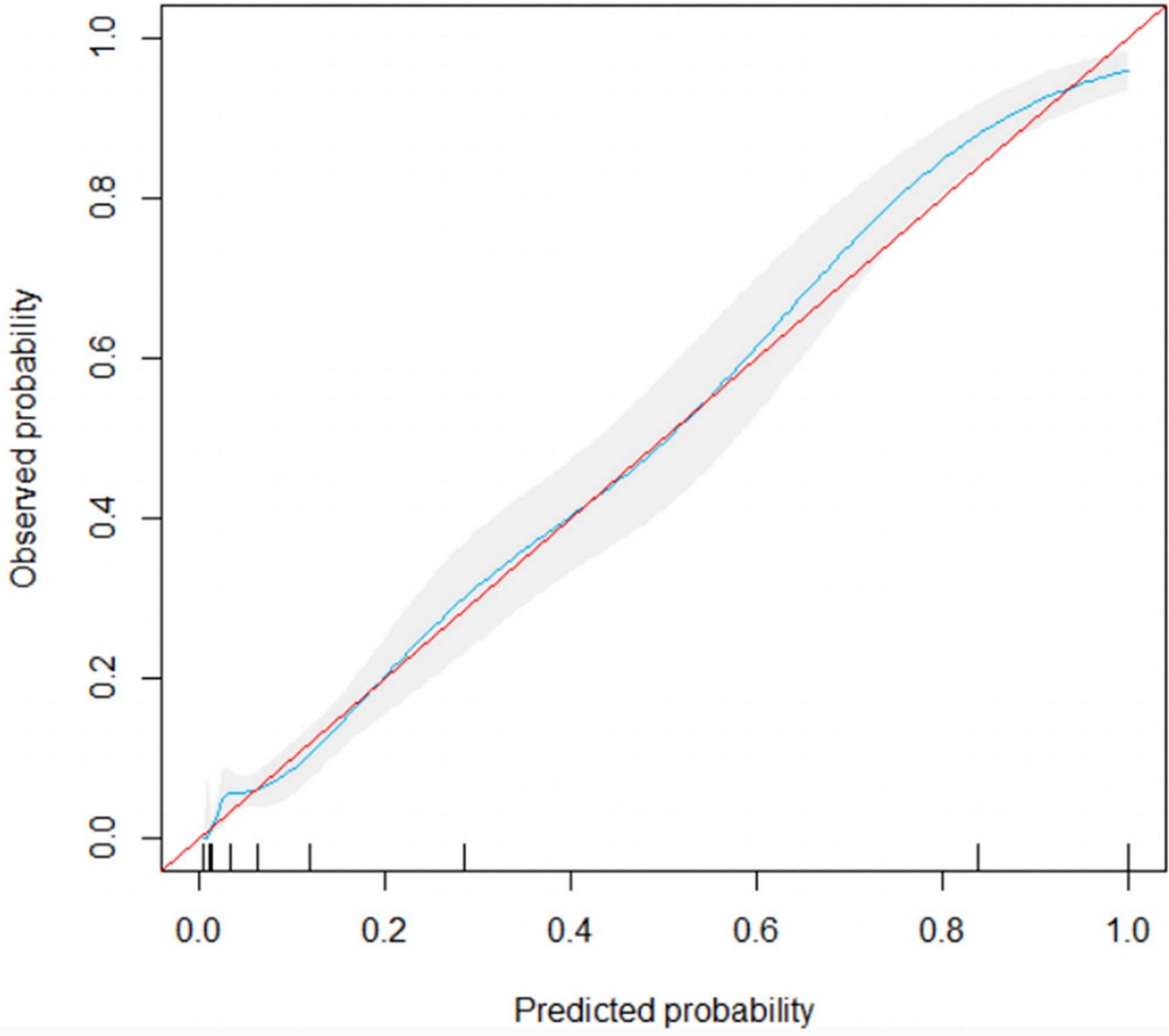

**Fig 3. Calibration curve comparing the probability of mortality predicted by the score and the probability of mortality observed in the patient population in the test set of data (slope = 1 and Brier score = 0.061).**

The estimates from the model may have the most value when triage decisions need to be made about which of patients to admit to a hospital or ICU bed, especially when the number of patients exceeds capacity. The model may be most useful for prioritizing patients at the extremes of prognosis. Notably, the 41% of patients in this cohort had scores of 0 had a very low probability of dying, and likely could have been cared for in outpatient settings, especially if periodic assessments of SpO2 and vital signs could be obtained. At the other extreme, over 90% of those with scores of 10 or more died, indicating a need to decide whether to implement or withhold aggressive treatment.

Our model may be currently useful in places outside of the US. The pandemic COVID-19 disease, hospitalization and death still continues to burden health systems in many countries, while abating in the U.S. Although our model was derived from the first wave of the pandemic

**Table 3. Total point score and risk of in-hospital death.**

| Total Score | Risk of Death % (95% C.I.) | N (%) |
|---|---|---|
| 0 | 0.8 (0.5–1.0) | 5677 (45.5) |
| 1 | 4.5 (3.5–5.5) | 1636 (13.1) |
| 3 | 9.7 (8.0–11.4) | 1137 (9.1) |
| 4 | 20.7 (17.6–23.8) | 936 (7.5) |
| 5 | 39.4 (34.6–44.1) | 404 (3.2) |
| 6 | 57.8 (51.9–63.8) | 268 (2.1) |
| 7 | 64.3 (58.6–69.9) | 277 (2.2) |
| 8 | 74.4 (69.2–79.7) | 266 (2.1) |
| 9 | 87.6 (82.8–92.4) | 185 (1.5) |
| 10 | 92.3 (88.6–95.9) | 207 (1.7) |
| 11 | 92.4 (88.2–96.6) | 157 (1.3) |
| ≥12 | 97.6 (96.5–98.8) | 674 (5.4) |

in New York City, the results and the model are likely to apply to other populations around the world. Our patient population is very racially diverse- Hispanic, Asian, Black and Caucasian, many of whom are recent immigrants and largely low income. We found that physiologic measurements of COVID-19 infection, such as low pSO2 and vital signs were very strong predictors of mortality while race, ethnicity did not influence outcome. New variants of the virus influence its transmission; they might cause more severe infection but are less likely to change the relationship between physiologic severity of the infection and risk of death. Any influence might mean that the model might underestimate the probability of death.

This analysis has several strengths. The algorithm was derived from a very diverse population of patients in New York City using data from 11 hospitals. The study population and number of deaths were large enough to produce estimates of mortality with narrow confidence intervals and high AUC values; it is unlikely that adding additional variables to the model would substantially improve its already high accuracy. Multivariable regression analysis of the variables selected by machine learning confirmed that they were strong and independent predictors of mortality. An easy-to-use version of the model is also universally available online for use in acute care settings danielevanslab.shinyapps.io/Covid_mortality/).

The analysis also has limitations. The model represents the natural history of COVID-19 disease before hospital care improved—and mortality rates declined—so it could not be calibrated to predict mortality with current standards of care. A large proportion of the patients who were admitted had low risk scores which reflects admission practices in NYC hospitals during the first wave of the pandemic. Although the study subjects included diverse races and ethnicities, we did not test the performance of this model in other study population. Further studies testing the performance of our model in other countries would be warranted. By design, the data did not include measurements, such as markers of inflammation and coagulation, or indices of comorbidity and severity of illness including presence of patients' symptoms, that predict mortality but that may not be readily available in the initial assessment of a patient. Thus, we did not calculate PSI, NEWS or CURB65 scores for comparison because our model used only data immediately available without referring to medical records.

## Conclusions

Mortality from COVID-19 illness can be rapidly and accurately predicted from a few vital signs that are readily available in acute care settings. When resources, such as hospital beds,

are scarce, estimates of the probability of dying might aid decisions about prioritizing patients to receive intensive care or other scarce resources. The prediction model, based on racially and ethnically diverse patients, is available online for use in clinical settings around the world.

## Supporting information

**S1 Fig. The sequence of boosted decision trees.** The first (top) figure (Tree 0) is the first boosted decision tree from the XGBoost model. The next (Tree 1) is the second boosted decision tree from the XGBoost model. The next tree (Tree 2) is the third boosted decision tree from the XGBoost model. The bottom tree (Tree 3) is the fourth boosted decision tree from the XGBoost model.
(TIF)

## Acknowledgments

Dr. Mitch Katz, CEO of NYC H+H, enabled the collaboration and access to NYC H+H data.

## Author Contributions

**Conceptualization:** Kyoung Min Kim, Daniel S. Evans, Jessica Jacobson, Warren Browner, Steven R. Cummings.

**Formal analysis:** Kyoung Min Kim, Daniel S. Evans, Jessica Jacobson, Xiaqing Jiang.

**Methodology:** Kyoung Min Kim, Daniel S. Evans, Steven R. Cummings.

**Project administration:** Warren Browner, Steven R. Cummings.

**Resources:** Jessica Jacobson.

**Supervision:** Steven R. Cummings.

**Writing – original draft:** Daniel S. Evans, Xiaqing Jiang, Warren Browner, Steven R. Cummings.

**Writing – review & editing:** Kyoung Min Kim, Jessica Jacobson, Xiaqing Jiang, Warren Browner, Steven R. Cummings.

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
