## [Decision Letter · Decision Letter 0]

7 Dec 2021

PONE-D-21-32900Rapid Prediction of In-hospital Mortality among Adults with COVID-19 diseasePLOS ONE

Dear Dr. Cummings,

Thank you for submitting your manuscript to PLOS ONE. After careful consideration, we feel that it has merit but does not fully meet PLOS ONE’s publication criteria as it currently stands. Therefore, we invite you to submit a revised version of the manuscript that addresses the points raised during the review process.

We look forward to receiving your revised manuscript.

Kind regards,

Chiara Lazzeri

Academic Editor

PLOS ONE

Journal Requirements:

Additional Editor Comments:

The topic is interesting and the study well designed and written. The Authors should hypothesized whether their model could be extended to noICU settings.

Reviewers' comments:

Reviewer's Responses to Questions

**Comments to the Author**

1. Is the manuscript technically sound, and do the data support the conclusions?

Reviewer #1: Yes

2. Has the statistical analysis been performed appropriately and rigorously? 

Reviewer #1: Yes

3. Have the authors made all data underlying the findings in their manuscript fully available?

Reviewer #1: Yes

4. Is the manuscript presented in an intelligible fashion and written in standard English?

Reviewer #1: Yes

5. Review Comments to the Author

Reviewer #1: Dear authors. Thank you so much for submitting this manuscript. Below, I am including some thoughts that hopefully would be useful to further strengthen your work.

-Line 62: When you mention rapidly available vital signs, could you clarify if you are referring exclusively to vital signs upon arrival to the hospital, or vital signs taken in any setting (i.e. at home, for example). It appears that Line 88 would suggest the former, but please clarify this point in Line 62 to satisfy the picky reader.

-Line 80: I appreciate very much the discussion of the motive behind your work. It is clear and it explicitly outlines the limitations of previous literature. Well done

-Introduction: Overall concise, well-motivated, and with a clearly stated objective. Again, well done

-Line 101: If relevant, could you clarify why you chose these these dates? Just curious !

-Line 119: Please briefly explain why the Kolmogrov-Smirnov test is the best statistical test for this context (this should be explained as a short clause only). Same with the Mann-Whitney U test. I think that it is not necessary to do the same for the Chi-squared test given that most readers should be familiar with it

-Line 134: Why a learning rate of 0.3? Why is this the best rate?

-Line 172: Please confirm that the 13,1910 patients were COVID positive, or COVID presumed patients, or all sorts of patients , just in the spirit of clarity

-Line 177: Do you have figures about the number of patients that were admitted for ICU-level of care, out of curiosity?

-Discussion: is there any data that you found on number of symptomatic days? For example, folks presenting with low BP and low SpO2 - I assume - likely had been symptomatic for more than days than say someone who has been experiencing symptoms for only one day. If relevant, please add this perspective to your discussion.

-You discuss the "easy to use" approach of your model very well. I am wondering: Could you describe how your model could be introduced to non-US settings? Are there further studies that would be needed for a low-income country, for example, to fully adopt your model? Or could any hospital abroad simply "start using" your model? If so, why do you think that is the case? I believe that you hint at all of this in your final paragraphs, but a richer Discussion on this end could be interesting.

-If you think that it would be relevant, I would encourage you to create a visual algorithm that could help a clinician triage their patients based on your model (similar to UpToDate models) - this would be a time-intensive case, but I strongly feel that visual algorithms could ultimately make it easier for others to better understand how to apply your model.

Some interesting papers that you may want to cite in your text, where you see fit:

-https://bmjopen.bmj.com/content/11/2/e045442.long

-https://onlinelibrary.wiley.com/doi/full/10.1002/rmv.2146

6. PLOS authors have the option to publish the peer review history of their article (what does this mean?). If published, this will include your full peer review and any attached files.

Reviewer #1: No

---

## [Author Response · Author response to Decision Letter 0]

16 May 2022

Reviewer #1: Dear authors. Thank you so much for submitting this manuscript. Below, I am including some thoughts that hopefully would be useful to further strengthen your work.

-Line 62: When you mention rapidly available vital signs, could you clarify if you are referring exclusively to vital signs upon arrival to the hospital, or vital signs taken in any setting (i.e. at home, for example). It appears that Line 88 would suggest the former, but please clarify this point in Line 62 to satisfy the picky reader.

Answer: We appreciate the insightful suggestions. In the present study, we aimed to develop a machine learning-based algorithm to rapidly predict the severity of COVID-19 infection, that is likely to lead to mortality, on or immediately after admission. Therefore, we used vital signs and parameters that were assessed upon arrival at the hospital. We tried to clarify more the time point when these predictors were collected as follows. 

Revised manuscript> 

-Line 63: Accurate prediction of the probability of death using rapidly available vital signs on arrival or immediately after admission to hospital without further testing of laboratory parameters or chest x-ray, might help prioritize patients for hospitalization, intensive care and intubation, or to receive limited treatments in places that have very limited resources.

-Line 80: I appreciate very much the discussion of the motive behind your work. It is clear and it explicitly outlines the limitations of previous literature. Well done

Answer: We appreciate the reviewer’s comments. Since the world still suffers from the COVID19 infection, several prediction models have been proposed from the diverse clinical settings or situations, in different study subjects, and using different predictors. Thus, prior to model development, we had thoroughly reviewed the performance, limitations, and strengths of other reported models. The strength of our model is that we only used readily available predictors in patients composed with diverse ethnicity. Therefore, we believe that we expect that our predictive model would outperform others especially in terms of generalizability and utility. We emphasized this point as the strengths of our model in the Discussion part. 

-Introduction: Overall concise, well-motivated, and with a clearly stated objective. Again, well done

Answer: Thank you for your positive response. 

-Line 101: If relevant, could you clarify why you chose these dates? Just curious !

Answer: In March 2020, the number of patients infected with COVID-19 in the United States started to increase dramatically, posing a serious health threat, and hospitals faced a shortage of beds for seriously ill patients. With this, we aimed to develop a model to predict the disease severity of COVID-19 infection for early classification of patients. So, in June 2020, we submitted an IRB and started analyses for this study. That is why these dates were finally set as a study period. 

-Line 119: Please briefly explain why the Kolmogrov-Smirnov test is the best statistical test for this context (this should be explained as a short clause only). Same with the Mann-Whitney U test. I think that it is not necessary to do the same for the Chi-squared test given that most readers should be familiar with it

Answer: There are several statistical methods to test normality, including the Shapiro-wilk test (SW), the Anderson-Darling test (AD), and the Kolmogrov-Smirnov test (KS). The Shapiro-wilk test is known to have the best statistical power at low sample sizes, but the statistical power becomes comparable among these methods at high sample sizes. Furthermore, SW test cannot be used with sample sizes greater than 5000. The statistics of KS test measures distance from the reference distribution and known to be a relative conservative test. It is also known to be sensitive of the two samples and can be applied larger sample size. Therefore, we chose the KS test as an alternative to test for normality. From the normality test, it turned out that most of predictors were not distributed normally, thus we used Mann-Whitney U test to compare baseline values between deceased subjects and survived subjects. we added this point why we used these methods for statistical analyses briefly as follows.

Revised manuscript> 

Line 120: The assumptions of normality for collected variables were tested Kolmogrov-Smirnov test, which is known to be sensitive in two samples. Baseline characteristics are presented median (IQR) for non-normally distributed continuous variables or N (%) for categorical variables. Because most of predictors were not distributed normally, the descriptive statistics for baseline characteristics were compared by in-hospital mortality using Mann-Whitney U test for continuous variables and Chi-squared test for categorical variables.

-Line 134: Why a learning rate of 0.3? Why is this the best rate?

Answer: We used the default value for the learning rate, which was 0.3. The learning rate (values range from 0 to 1) is a factor that shrinks the feature weights after each boosting step. Higher values have very little shrinkage and result in models being fit with fewer boosting steps. Lower values have more shrinkage and require more boosting steps and more computing time. While we could have experimented with this parameter to minimize the computing time required to fit a well-performing model, we were quite satisfied with the default learning rate that achieved a short computing time (<10 seconds) to fit a model with such good performance in the test dataset (AUC 94%). We didn’t want to search out a model with unusual parameter settings just to improve performance by a few percent. Thus, we left the learning rate at the default value.

-Line 172: Please confirm that the 13,190 patients were COVID positive, or COVID presumed patients, or all sorts of patients , just in the spirit of clarity.

Answer: All the 13,190 patients were confirmed as COVID19 infection. We edited the sentence as follows.

Revised manuscript> 

Line 175: Between March 1 and June 30, 2020, 13,190 patients who confirmed with COVID-19 infection, were admitted to a NYC H+H hospital.

-Line 177: Do you have figures about the number of patients that were admitted for ICU-level of care, out of curiosity?

Answer: Among 13,190 subjects, 2,227 (16.9%) patients were cared in the ICU during the hospitalizations. As expected, the subjects who cared in the ICU had a higher mortality rate than those who did not. We added this number in the manuscript since it helps to figure the general severity of the patients included in this study.

Revised manuscript> 

Line 176: Among them, 2,227 (16.9%) patients were cared in ICU during hospitalizations.

-Discussion: is there any data that you found on number of symptomatic days? For example, folks presenting with low BP and low SpO2 - I assume - likely had been symptomatic for more than days than say someone who has been experiencing symptoms for only one day. If relevant, please add this perspective to your discussion.

Answer: We appreciate the reviewer’s valuable comment. However, unfortunately, we did not assess any parameters about subjective symptoms (presence, severity, or duration of symptom) in this study. Although the presence, duration or severity of symptoms is likely associated with a poorer prognosis for patients, the purpose of this study is to establish a predictive model just using objective and readily measurable parameters. Therefore, we did not include parameters related to the patient's symptoms in this analysis. We added this point as a limitation of the study as follows;

Revised manuscript> 

Line 279: By design, the data did not include measurements, such as markers of inflammation and coagulation, or indices of comorbidity and severity of illness including presence of patients’ symptoms, that predict mortality but that may not be readily available in the initial assessment of a patient.

-You discuss the "easy to use" approach of your model very well. I am wondering: Could you describe how your model could be introduced to non-US settings? Are there further studies that would be needed for a low-income country, for example, to fully adopt your model? Or could any hospital abroad simply "start using" your model? If so, why do you think that is the case? I believe that you hint at all of this in your final paragraphs, but a richer Discussion on this end could be interesting.

Answer: As we described in the Discussion part, the study subjects in the present study composited from diverse ethnicity. Therefore, we cautiously expect this model would work in other ethnic groups as well. However, we did not test the performance of this model in other populations or non-US setting, so we cannot conclude yet. Therefore, further studies testing the performance of our model in other countries would be warranted. We added this point in the discussion as follows. 

Revised manuscript> 

Line 277: Although the study subjects included diverse races and ethnicities, we did not test the performance of this model in other study populations. Further studies testing the performance of our model in other countries will be warranted.

-If you think that it would be relevant, I encourage you to create a visual algorithm that could help a clinician triage their patients based on your model (similar to UpToDate models) - this would be a time-intensive case, but I strongly feel that visual algorithms could ultimately make it easier for others to better understand how to apply your model.

Answer: Thank you for the suggestion. We additionally provided the four decision trees of our model in supplemental figure in the revised manuscript. However, assessing an individual with four decision trees is not very easy. Thus, we also provide our web application that visualizes the risk category a person would fall into with values of different variables.

Revised manuscript> 

Supplemental Figure 1. First boosted decision tree from XGBoost model. 

Some interesting papers that you may want to cite in your text, where you see fit:

-https://bmjopen.bmj.com/content/11/2/e045442.long

-https://onlinelibrary.wiley.com/doi/full/10.1002/rmv.2146

Answer: We appreciate the reviewer’s suggestion and we additionally cited these references in the manuscript as new reference 1 and 23.

---

## [Editor Report · Decision Letter 1]

30 May 2022

Rapid Prediction of In-hospital Mortality among Adults with COVID-19 disease

PONE-D-21-32900R1

Dear Dr. Cummings,

We’re pleased to inform you that your manuscript has been judged scientifically suitable for publication and will be formally accepted for publication once it meets all outstanding technical requirements.

Kind regards,

Chiara Lazzeri

Academic Editor

PLOS ONE
---

## [Editor Report · Acceptance letter]

7 Jul 2022

PONE-D-21-32900R1 

Rapid Prediction of In-hospital Mortality among Adults with COVID-19 disease 

Dear Dr. Cummings:

I'm pleased to inform you that your manuscript has been deemed suitable for publication in PLOS ONE. Congratulations! Your manuscript is now with our production department. 

Kind regards, 

on behalf of

Dr. Chiara Lazzeri 

Academic Editor

PLOS ONE